# Evaluation of the Efficacy of Different Irrigation Systems on the Removal of Root Canal Smear Layer: A Scanning Electron Microscopic Study

Vincenzo Tosco [1], Riccardo Monterubbianesi [1], José Aranguren [2], Lucia Memè [1], Angelo Putignano [1] and Giovanna Orsini [1,*]

[1]   Department of Clinical Sciences and Stomatology (DISCO), Università Politecnica delle Marche, 60126 Ancona, Italy
[2]   School of Dentistry, Universidad Rey Juan Carlos (URJC), Av. Atenas, S/N, 28922 Alcorcon, Spain
*    Correspondence: g.orsini@univpm.it

**Abstract:** Irrigation represents a crucial step in endodontics for bacteria disinfection and smear layer removal. Several irrigation strategies have been proposed, although their effects are controversial. This study aims to assess the comparison of four different irrigation systems on the smear layer removal utilizing scanning electron microscopy (SEM). Forty sound monoradicular teeth were collected and casually allocated into four groups ($n = 10$): Group A, conventional irrigation; Group B, IrriFlex irrigation; Group C, ultrasonic irrigation system; Group D, apical negative pressure irrigation. After chemo-mechanical preparation and longitudinal root separation, the three root thirds were analyzed by SEM. Micrographs were carried out at $\times 2000$ to analyze the smear layer residual in each third of the root canal. Statistically significant differences were found between Group A and Group D ($p < 0.05$). Groups B and C showed satisfactory results in the coronal and middle thirds, while Group D was the only system that achieved satisfactory results in the apical third. However, the complete smear layer removal in the root canal was never achieved with any of the four systems tested. In conclusion, among the four tested irrigation systems, the greatest efficacy on smear layer removal was reached by Group D.

**Keywords:** endodontics; irrigation; smear layer; disinfection; dental material





## 1. Introduction

The main goal in endodontics is the complete removal of bacteria biofilms and their by-products from the root canal system and, in doing so, preventing consequent intracanal contamination. The success of endodontic treatment derives from meticulous root canal preparation and disinfection, an adequate filling and a proper coronal and apical seal. Although current technology offers several instruments capable of performing excellent root canal preparation [1] and new nanomaterials which guarantee an appropriate apical sealing, several problems persist, including treating teeth with complex root canal morphology, completely eliminating bacteria and the smear layer removal. Indeed, the different irregularities in the root canal morphology, such as oval and lateral canals, isthmus and apical deltas [2], hinder the mechanical instrumentation and, as a result, do not ensure the complete disinfection of bacteria [3], leaving approximately 35–53% of the root canal walls unprepared [4,5]. In addition, the mechanical instrumentation produces an unstructured mass called a smear layer, which is accumulated on the wall of root canals. The smear layer is composed of inorganic and organic material [6]. Its removal is crucial as the smear layer prevents the infiltration of the irrigant solutions and sealants inside the dentinal tubules and hinders the correct disinfection and sealing of the canal system. Therefore, residual bacteria and smear layer decrease the system tightness [7], prevent the complete disinfection of the root canal [8] and reduce the filling material adhesion, leading to poor

long-term prognosis [9]. In this light, in order to improve the efficacy of root canal disinfection procedures, it is necessary to combine mechanical instrumentation with the chemical action of the irrigants [10,11]. Indeed, this so-called chemo-mechanical preparation allows the irrigant solutions to act directly on the walls of the root canal and penetrate inside the dentinal tubules, considerably reducing the bacterial load [12]. Sodium hypochlorite (NaOCl) has been demonstrated to be the irrigating agent of choice for its antibacterial properties and its ability to dissolve necrotic tissues [13–15], despite its high toxicity [16] and inability to remove the inorganic component of the smear layer [12]. Therefore, NaOCl is usually combined with a chelating solution such as ethylenediaminetetraacetic acid (EDTA), which is effective in dissolving the smear layer from the inner surfaces of the root canal [17,18]. However, the typical irrigants are administered into the root canal space using a conventional needle which has limitations related to the needle tip insertion [19] and their incapability to cleanse areas not prepared by mechanical instrumentation [20,21]. Therefore, different systems and techniques have been provided to increase and facilitate the infiltration of the irrigant solution along the root canal system, with promising results. One of the most used techniques for intra-canal irrigation is the ultrasonic activation of the irrigant solution through a file or a vibrating tip inside the canal, whereby the solution can be injected intermittently or continuously [22,23]. Another technique suggests preheating NaOCl solution as it more efficiently dissolves pulp tissue and cleans the canal [24]. Moreover, EndoVac (Discus Dental, Culver City, CA, USA), an apical negative pressure (ANP) irrigation system, is a safe dispensing system designed to allow irrigants to penetrate all inaccessible areas of the root canal system [25], preventing the extrusion of NaOCl out of the apex and hence reducing the probability of accidents [16,26]. Several studies highlight the high performance of these two irrigation systems in regard to the delivery and penetration of the irrigating solution up to the most difficult areas of the root canal [27–29]. Despite these results, it is still debated in the literature which technique is the most effective in removing the smear layer [30,31]. Therefore, several new tools and techniques are provided to improve canal cleaning, such as IrriFlex (Produits Dentaires SA, Veyvey, Switzerland), a thin, flexible polypropylene tip, and Z-Activator (Zarc system, United Dental Changzhou, Changzhou, Jiangsu, China), an ultrasonic system that uses high frequency to activate the irrigation solution, although these techniques have not yet been thoroughly investigated. For this reason, the purpose of this study was to analyze different irrigation techniques, including the two new tools on the market, to evaluate their efficacy on the smear layer removal from root canal walls using a Scanning Electron Microscopy (SEM).

The null hypothesis of this study is that there are no differences in smear layer removal among the different irrigation systems analyzed.

## 2. Materials and Methods

### 2.1. Specimen Preparation

A total of 40 single-root sound lower premolars from individuals aged 18 to 35 years were used. These teeth were extracted for periodontal and orthodontic causes, according to the Helsinki Declaration of 1964 and the guidelines of the Local Ethics Committee. Therefore, as a waste product from surgical procedures, these hard dental tissue samples were used for research purposes, subject to the patients' informed consent. The exclusion criteria were the presence of root caries, fractures or cracks, previous endodontic treatment; only teeth with an intact and mature apex and roots longer than 15 mm were selected. Exclusion criteria were detected using a 10x objective (Eclipse Ni, Nikon, Amstelveen, The Netherlands). Furthermore, the teeth with apical curvature greater than 30° (according to the Schneider method [32]) were visualized by buccolingual and mesiodistal X-ray analysis and then discarded. On the basis of the radiographic evaluation, the dental elements with anatomy responding to Vertucci's classification in Class 1 [33] and as similar to each other as possible were selected. In the selected teeth, connective tissue residuals and debris were eliminated with hand-scaling tools and stored in chloramine solution (NH2Cl) in 0.5% $w/w$ and at ambient temperature. One single expert endodontist performed all procedures to



avoid operator bias. The samples were decoronated with a diamond disc (Komet Dental, Lemgo, Germany) mounted on a surgical handpiece (Kavo Dental, Biberach an der Riss, Germany) to obtain a 16 mm standardized root length. A varnish was applied on the outer surface of the apex and root of each specimen to avoid the extrusion of irrigants out of the apex [17].

### 2.2. Root Canal Instrumentation

All samples were treated with the same mechanical instrumentation system. Pro-Train device (Simit Dental, Mantova, Italy) was used to fix samples during every root canal instrumentation. A 10 K-file (Dentsply Maillefer, Ballaigues, Switzerland) was inserted into the root canal, and the working length (WL) was established electronically using an apex locator (Root ZX, Morita, Osaka, Japan). The glide path was performed with the Z1 file of BlueShaper Zarc (Zarc4Endo, Shenzhen Denco Medical CO., LTD, Xiawei Industrial Zone, Shenzhen, China). Then, the specimen was instrumented using the remaining BlueShaper Zarc files following the sequence according to the manufacturer's recommendation: Z2, Z3 and Z4 files at 500 rpm and 4 Ncm (speed and torque, respectively). At every step of instrumentation, the irrigation was performed with 5.25% NaOCl using a 30-gauge closed-end needle polypropylene tip (IrriFlex, Produits Dentaires SA, Veyvey, Switzerland) by introducing it into the canal at 2 mm from the apex. Prior to the irrigation tested protocols, the root canal was irrigated for 1 min with 3 mL of 5.25% NaOCl, 1 mL of 17% EDTA (OGNA Lab S.r.l., Muggiò, MB, Italy) followed by 3 mL of 5.25% NaOCl [34].

Afterwards, samples were arbitrarily allocated into four groups ($n$ = 10) according to the irrigation system, as described below.

**Group A.** Conventional needle irrigation technique that distributes the solution with a needle inside the canal. The irrigation was provided with 10 mL of NaOCl at 5.25% (Niclor 5, Ogna; Italia) through the Z-Rinse needle (United Dental Changzhou, Changzhou, Jiangsu, China) at a rate of 1 mL/min. The teeth were rinsed with 3 mL of physiological solution, and a wash with 3 mL of liquid EDTA at 17% for 1 min. was performed.

**Group B.** IrriFlex irrigation. The irrigation was provided with 10 mL of NaOCl at 5.25% (Niclor 5, Ogna; Italia) through the IrriFlex (Produits Dentaires SA, Veyvey, Switzerland) into the canal with a dispensing time of 1 mL/min. The teeth were rinsed with 3 mL of physiological solution, and a wash with 3 mL of liquid EDTA at 17% for 1 min. was performed.

**Group C.** Ultrasonic activation system. The irrigation was performed by inserting 10 mL of NaOCl at 5.25% (Niclor 5, Ogna; Italia) using IrriFlex (Produits Dentaires SA, Veyvey, Switzerland) plus activation with Z-Activator Zarc system (United Dental Changzhou, Changzhou, Jiangsu, China) at the maximum power of 45 kHz. The size 20.04 Z-silver tip was used to avoid rubbing against the canal, always keeping a distance of 2–3 mm from WL. The Z-Activator device was used with short up-and-down vertical movement with an oscillation of 2–3 mm for thirty seconds. This process was repeated ten times (30 s each), and at all steps, the solution and debris were suctioned, and the canal was filled with the new irrigant (1 mL NaOCl each).

**Group D.** The ANP irrigation system was performed using EndoVac delivering a total of 5 mL of NaOCl at 5.25%. The master delivery tip (MDT) was inserted into the coronal third for the delivery of 1 mL of NaOCl at 5.25%, after which the suction macro-cannula was applied progressively inside the canal in up-and-down movements for 30 s. Then, with the same approach, the micro-cannula was placed 1 mm from the WL inserting a 1 mL of NaOCl at 5.25% for 1 min, again using up-and-down movements for 1–2 mm [26]. The procedure was repeated using 3 mL of 17% EDTA for 1 min to uncover the dentinal tubules from the smear layer. The last step was carried out with a 3 mL NaOCl at 5.25% irrigation with the micro-cannula aspiration for 1 min.

Finally, all root canal samples were rinsed for 30 s using ethanol and dried with Z4 paper points (Zarc4Endo, Shenzhen Denco Medical CO., LTD, Xiawei Industrial Zone, Shenzhen, China). After that, the samples were prepared for SEM visualization. A schematic diagram of the experimental setup is shown in Figure 1.

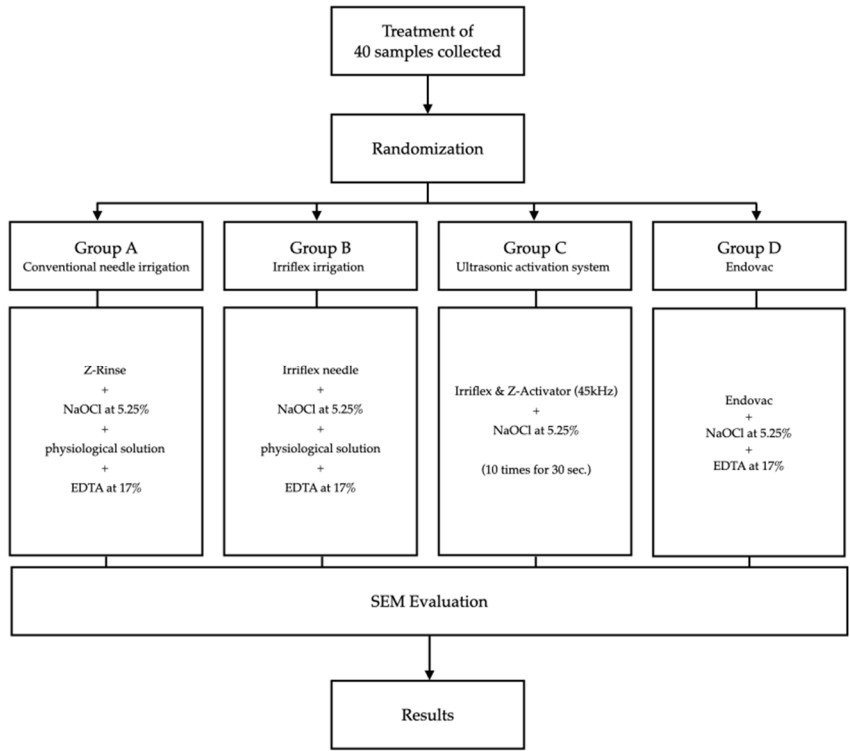

**Figure 1.** A schematic diagram of the experimental setup.

*2.3. SEM Investigation*

SEM analysis aimed to analyze the presence of the smear layer on the internal surface of each root. The samples were fixed for 4 h in 2.5% glutaraldehyde in 0.1 M cacodylate buffer at pH 7.4, according to our previous study [35]. Then, they were crown-apically split using a low-speed diamond saw (Buehler Isomet Low Speed Saw) and smoothened with a lapping machine (Buehler Metaserv, Buehler, Lake Bluff, IL, USA) using 600- and 800-grit silicon carbide abrasive papers under constant deionized water irrigation in order to obtain a superficial flat surface which was used to lightly groove the three thirds of the root (coronal, middle and apical regions) to standardize SEM analysis [36]. A Z4 gutta-percha calibrated cone (Zarc4Endo, Shenzhen Denco Medical Co., Ltd, Xiawei Industrial Zone, Shenzhen, China) was placed inside each sample to avoid canal contamination during the root section [34]. Finally, specimens were gold sputter-coated, air-dried and placed on an aluminium holder for SEM observation. Scanning electron micrographs were taken by a field emission electron microscope Zeiss Supra 40 operating at 30 kV and with a working distance of 12 mm, using magnifications of 2000X [37]. Blindly, two independent expert operators assessed the presence or absence of smear layer and debris on the coronal, middle and apical thirds of the root canal surface. The evaluation system applied follows the Hülsmann et al. criteria [38] for results validation, as reported:

- Score 1: Absence of smear layer, orifices of dentinal tubules uncovered;
- Score 2: A small quantity of smear layer, some dentinal tubules open;
- Score 3: Uniform smear layer covering the root canal walls, scarce dentinal tubules uncovered;
- Score 4: Whole root canal wall was hidden by a uniform smear layer; no dentinal tubules uncovered;
- Score 5: Substantial, uniform smear layer covering the whole root canal walls.

Finally, the scores obtained were used for statistical analysis to evaluate the removal of the smear layer for each third of the root of the four groups tested.

### 2.4. Statistical Analysis

Data were analyzed by nonparametric tests. Comparison of smear layer and debris score in experimental groups were analyzed by Kruskal–Wallis and Mann–Whitney tests for the intra and inter-group comparison of mean smear layer removal at the apical, middle and coronal thirds. The statistical software Prism9 (GraphPad Software, Inc., San Diego, CA, USA) was used for all the analysis. The sample size for each group was determined to achieve a power of 0.8 and significance level of 0.05.

## 3. Results

Results obtained from micrographs scoring are presented in detail in Table 1. The mean scores of the smear layer and the intergroup and intragroup comparisons are presented in Table 2.

**Table 1.** Numerical value and percentage scores (%) of smear layer in agreement with Hülsmann et al. criteria per group.

| Tested Group | Root Canal Third | Score #1 | Score #2 | Score #3 | Score #4 | Score #5 |
|---|---|---|---|---|---|---|
| Group A | Coronal | 0 | 15 (50%) | 14 (46.7%) | 1 (3.3%) | 0 |
| | Middle | 0 | 0 | 19 (63.3%) | 6 (20%) | 5 (16.7%) |
| | Apical | 0 | 0 | 8 (26.7%) | 9 (30%) | 13 (43.3%) |
| Group B | Coronal | 17 (56.7%) | 13 (43.3%) | 0 | 0 | 0 |
| | Middle | 0 | 19 (63.3%) | 6 (20%) | 5 (16.7%) | 0 |
| | Apical | 0 | 0 | 13 (43.3%) | 10 (33.3%) | 7 (23.4%) |
| Group C | Coronal | 17 (56.7%) | 8 (26.7%) | 5 (16.6%) | 0 | 0 |
| | Middle | 0 | 15 (50%) | 10 (40%) | 5 (10%) | 0 |
| | Apical | 0 | 0 | 21 (70%) | 5 (16.7%) | 4 (13.3%) |
| Group D | Coronal | 23 (76.7%) | 7 (23.3%) | 0 | 0 | 0 |
| | Middle | 16 (53.3%) | 14 (46.7%) | 0 | 0 | 0 |
| | Apical | 0 | 15 (50%) | 11 (36.7%) | 4 (13.3%) | 0 |

**Table 2.** Mean values of smear layer scores (mean ± standard deviation) in different groups (A, B, C and D) divided by the three regions of root canal (Coronal, Middle and Apical).

| | Group A | Group B | Group C | Group D |
|---|---|---|---|---|
| Coronal | 2.53 ± 0.57 [a,1] | 1.43 ± 0.5 [b,1] | 1.6 ± 0.77 [b,1] | 1.23 ± 0.43 [b,1] |
| Middle | 3.53 ± 0.78 [a,2] | 2.53 ± 0.78 [b,2] | 2.67 ± 0.76 [b,2] | 1.47 ± 0.51 [c,1] |
| Apical | 4.17 ± 0.83 [a,3] | 3.8 ± 0.81 [ab,3] | 3.43 ± 0.73 [b,3] | 2.63 ± 0.72 [c,2] |

Each value was compared intra and between groups by using Kruskal–Wallis and Mann–Whitney tests ($p < 0.05$). Different superscript letters (a, b, c) indicate significant differences between groups within the same row. Different superscript numbers (1, 2, 3) indicate significant differences between the regions of root canal within the same column.

Group A and Group D present significant differences ($p < 0.05$) for smear layer removal score, as reported in Figure 2, which describes the smear layer reduction after the tested irrigation methods in each of the root canal thirds. The apical third of Group A displayed the highest percentage of Score #5, while Group D showed the lowest score for the apical third (Score #2).

These findings are displayed in Figure 3, which presents the morphology of different root canal wall surfaces. The representative scanning electron micrographs of Group A showed several opened dentinal tubules with the presence of a small amount of smear layer in the coronal third (Score #2). In contrast, the middle third displayed a greater number of closed dentinal tubules due to the growing presence of the smear layer, which instead completely covered the wall surface of the apical third (Score #5). The irrigation system

proposed in Group B and Group C showed better cleaning and removal ability of the smear layer than that of Group A. Indeed, for both groups, the coronal third displayed opened dentinal tubules without smear layer residuals (Score #1) and Score #2 in the middle third.

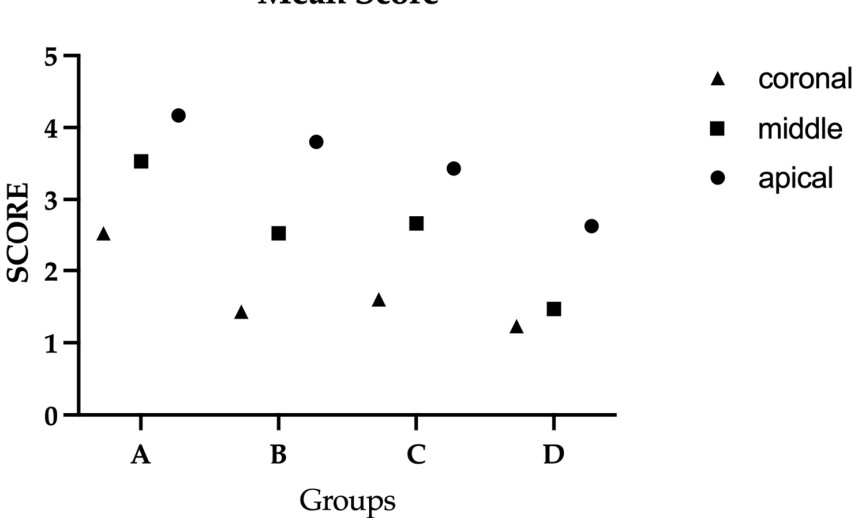

**Figure 2.** Mean scores graphic of the smear layer for the coronal, middle and apical third of the canals.

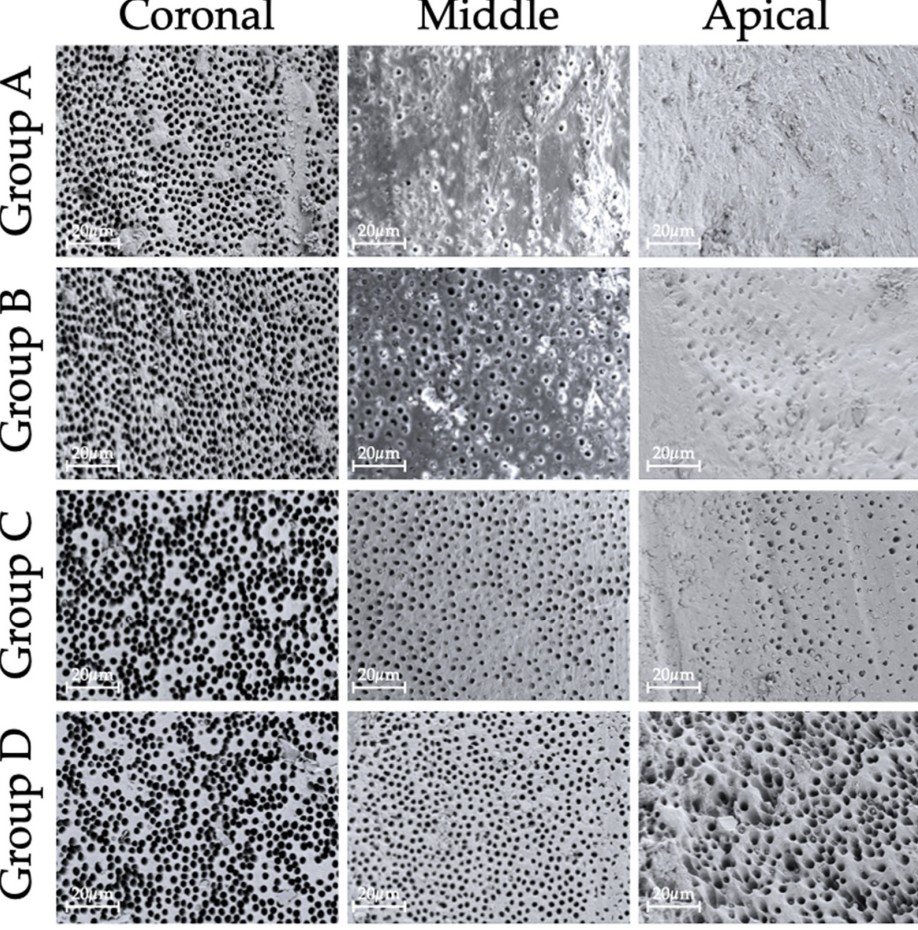

**Figure 3.** Representative scanning electron micrographs of the three canal thirds (coronal, middle and apical) of every irrigation system tested (2000X mag., 30 KV accelerated voltage).

In addition, for the apical third, both groups presented a uniform smear layer hiding the root canal surface with scarce empty dentinal tubules (Score #3). Finally, optimal results were obtained by Group D, which highlights the lack of smear layer in both the coronal and middle third and a Score #2 for the apical third.

## 4. Discussion

Shaping and cleaning of the root canal represents an important step in endodontics to obtain bacterial load reduction and ensure long-term healing. Several studies demonstrated that mechanical instrumentation alone is unsuccessful in completely removing the remaining bacteria and smear layer [2,5,8]. Irrigation plays a crucial role during endodontic treatment as it removes the smear layer and bacteria and decontaminates the surfaces of the root canal system, even when involving anatomical complexities [23,39]. Several protocols and systems have been introduced that combine irrigating solutions and activation methods in order to achieve satisfactory disinfection. They decrease the side effects of irrigants on dentine structure and increase the capability of endodontic and restorative materials in sealing and filling the root canal system [40]. This work aimed to investigate the efficacy of four different irrigation protocols for removing the smear layer. Since some groups present statistically significant differences, the null hypothesis is partially rejected. Indeed, Group A demonstrates the capability of the conventional needle irrigation to obtain good results in the coronal root third with a Score #1; however, in the remaining two-thirds, the removal of the smear layer appears insufficient. Indeed, scanning electron micrographs show a massive presence of a smear layer which completely covers the wall of the apical third, as indicated by Score #5 according to the criteria of Hülsmann et al. [38]. This finding is in agreement with several research studies, which indicate the insufficient ability of the conventional needle system to remove the smear layer in the apical third [41–43]. Furthermore, the obtained results demonstrate that the amount of smear layer remaining in the coronal and middle levels is less than that of the apical third in all four groups, thus proving that less irrigation is delivered to the apical region due to the difficulty in reaching a narrower space. In addition, the diminished performance of the conventional irrigation system in the apical region can be attributed to the formation of the vapor lock or, rather, an air bubble or voids formation inside the canal, which avoids the proper irrigant delivery to the apical third [44]. Contrarily, the other three tested protocols show the capability to eliminate the vapor lock, allowing the access of irrigants to the apical area, as displayed in Figure 3. Furthermore, it is noteworthy that the IrriFlex propylene tip tested in Group B allows the irrigant solution to reach and clean the apical area, even in anatomically complex areas, due to the flexibility and 30G needle shape. Although it is not evident in this study, the literature claims that there is a need to use other protocols and instruments to activate the irrigation solution and obtain adequate disinfection and cleaning of the entire endodontic space. The ultrasonic irrigation is probably one of the most generally used strategies to optimize disinfection [2]. Indeed, the activation system tested in Group C displays promising results, although a uniform smear layer still covers the root canal surface in the apical third. Many investigations underlined that ultrasonic activation methods of irrigants are necessary to produce clinical effectiveness [45–48]. These systems exploit the cavitation effect, which consists of the process in which millions of bubbles are formed during canal irrigation [10]. These bubbles grow in size, move around, and burst, significantly amplifying the cleaning effect of the system. The movement of the tip produces a special effect called acoustic turbulence, which accelerates the movement of the liquid inside the root canal [49]. Ultrasonic agitation suspends tissue remnants and dentin debris, transporting them coronally and making their removal easier [50]. It also helps irrigants to reach the entire anatomy of the canal, including isthmuses, lateral canals, apical ramifications, etc. For this reason, also in this work, the effective rule of ultrasonic activation demonstrates a better result in smear layer removal than the conventional system. Furthermore, this is the first study that evaluates a new ultrasonic device just released on the market, providing the energy for tip oscillation

and vibration in high frequency (45 kHz $\pm$ 5 kHz), as required to create sufficient acoustic streaming and cavitation. However, the apical third still presents a certain amount of smear layer. Therefore, the disinfection and cleaning of this area are rather inconclusive, as also highlighted by the existing scientific literature [5,51,52]. Conversely, the ANP irrigation protocol studied in Group D shows the best results in all thirds of the root, as supported by several studies [30,53,54]. Indeed, the EndoVac system provides a clean surface without debris and a small amount of smear layer and opened dentin tubules down to the apical third of the root canal due to the apical current flow generated by the apical negative pressure. Since the irrigant comes in direct contact with the entire dentinal walls taking the irrigant to the full WL, the results of this work support the literature showing the great ability of EndoVac. Nonetheless, the excellent performance of this ANP device is debated in the literature, as many studies showed no statistically significant differences between ANP and ultrasonic activation [34,55,56]. This result can be due to the lower concentration of the irrigant solution used (0.5% versus 5.25% applied in this study) [55] and to the evaluation of bacteria disinfection of the overall root canal without considering the smear layer elimination in the apical region [56]. Noteworthy is the fact that many works in the literature are in vitro studies using SEM analysis. SEM is an excellent method for this investigation, although it presents several limitations, such as the fact that samples must be sectioned for analysis, and that it is a subjective qualitative evaluation that can be quantified by several scoring systems [38,55,57]. Furthermore, since SEM analysis alone does not allow a correct longitudinal observation of the dentin morphology, it is necessary to complete the evaluation with a Micro-CT analysis which is more reliable and therefore represents a fundamental requirement for sample selection and for studying the smear layer removal procedures [27,58]. Another possible limitation of this work is the complex anatomy of the root canal system, which makes the complete cleaning of the superficial root canal walls difficult. Further in vitro and in vivo studies are needed to validate these obtained results, so the tested protocols can help the clinician achieve success in endodontic treatment.

## 5. Conclusions

The ultrasonic activation system used with a thin polypropylene needle shows satisfactory results in the coronal and middle thirds, while the ANP system is the only irrigation system that achieves satisfactory results on smear layer removal in the apical third. Nevertheless, its total elimination from the apical third of the root canal still poses a real challenge. Further studies are needed to better understand the behavior of the new irrigation systems.

**Author Contributions:** Conceptualization, V.T., G.O. and A.P.; methodology, V.T. and R.M.; software, R.M.; validation, J.A. and A.P.; formal analysis, J.A. and L.M.; investigation, V.T. and R.M; resources, A.P.; data curation, V.T. and R.M.; writing—original draft preparation, V.T. and R.M.; writing—review and editing, J.A., L.M. and G.O.; visualization, V.T. and R.M.; supervision, G.O. and A.P.; project administration, A.P.; funding acquisition, A.P. All authors have read and agreed to the published version of the manuscript.

**Funding:** This research received no external funding.

**Institutional Review Board Statement:** The study was conducted in accordance with the Declaration of Helsinki.

**Informed Consent Statement:** Written informed consent has been obtained from the patient(s) to publish this paper.

**Data Availability Statement:** Not applicable.

**Acknowledgments:** All authors extend their acknowledgments to Andrell Hosein for English style edition and to Adriano Di Cristoforo for his valuable support in SEM observations.

**Conflicts of Interest:** The authors declare no conflict of interest.

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
