# Peer review of "Evaluation of the Efficacy of Different Irrigation Systems on the Removal of Root Canal Smear Layer: A Scanning Electron Microscopic Study"

_applsci, doi:10.3390/app13010149_

Round 1

Reviewer 1 Report

In introduction please indicate the study null hypothesis and also the novelty of the current study.

In methods section add the sample size calculation.

Table 1- I consider adequate to indicate the numerical value and in barckets the percentage and also add a supplemetary colum for the p-value in each cathegory.

Author Response

Dear Reviewer, thank you for the time and efforts dedicated to providing your valuable feedback to our Ms. By following Reviewer’ suggestion, we have corrected and improved the Ms, responding point-by-point to your comments and concerns. The changes have been highlighted in red within the Ms.

  1. In introduction please indicate the study null hypothesis and also the novelty of the current study.

In the Introduction section, we added the null hypothesis and the novelty of this work (see lines 73-86).

  1. In methods section add the sample size calculation.

Sample size was based on qualitative variables. Moreover, the sample size used was the one calculated for previous studies [30,38] see lines 193-194.

  1. Table 1- I consider adequate to indicate the numerical value and in brackets the percentage and also add a supplementary column for the p-value in each category.

The Table 1 was improved with the numerical value and, in brackets the percentage. Moreover, Table 2 was added to better describe the statistical results for both intergroup and intragroup, using letters and numbers to specify the differences.

Reviewer 2 Report

This manuscript constitutes an attempt to evaluate the efficacy of three different irrigating systems in removal of intra canal smear layer by conducting an ex-vivo study. The study is relevant to the journal. However, there is no novelty in this study. Previous similar studies have been published in endodontic literature (J Endod 2015;41:1660–1666). Also, the methodology employed is not appropriate. Manuscript is very poorly written. English needs to be improved significantly. Some of the specific queries have been addressed below.

*Introduction:

- Mention previous studies on the efficacy of passive ultrasonics and Endovac in removal of smear layer from root canal system.

- Mention the hypothesis of the study tested.

*Methodology:

- How was the sample size estimated?

- What was the age groups of patients from whom the teeth were collected? This is one of the very important confounding factor which can influence the results because, in elder patients, the apical portion of the root canal system will be calcified without any open dentinal tubules. This may skew the results when the samples are examined under SEM.

- Authors should provide detailed information regarding how samples were selected, anatomically matched and how groups were formed – authors stated that “single-rooted human teeth” were used. But which single-root teeth? Which canal anatomy? Circular, oval? Nowadays there are several methods that allows authors have similar anatomy and this has to be perfomed otherwise the results are not valid.

- Mention the ethical approval number of the project.

- How was exclusion criteria’s evaluated?

- The volume of the irrigant should be mentioned as “mL” and not “cl”.

- It is mentioned that, samples were allocated to three groups (n=10). However, four groups A, B, C, D are mentioned. Clarify!

- Randomization should have been performed during the allocation of the samples into different groups to avoid bias.

- In groups A and B, what was the time duration of irrigation with NaOCl and EDTA?

- Why two different types of needle irrigation was performed (Z Rinse and Irrifex)?

- Mention the manufacturer details of EDTA.

- In group C, what was the time period of NaOCl irrigation? Also, it is not clear in between every cycle of ultrasonic activation, NaOCl was used or not. Mention in detail.

- What was the power setting of ultrasonic device used?

- In group D, mention the time duration of irrigation with NaOCl and EDTA.

- In SEM analysis, why the specimens were finished and polished after longitudinal sectioning of the root? What was the rationale behind this?

- Knowing that our instruments touch only 50% of the canal walls  (J Endod 2011; 37:13947; J Endod 2015;41:1545– 50), what is the guarantee that the areas imaged in SEM had a smear layer to begin with?

- The methodology used to assess smear layer using SEM model is not a sound, reproducible and valid one. Ideally, a longitudinal observation of the canal using micro-CT is regarded a fundamental requirement to study the smear layer removal procedures. Thus, the scanning electron microscopic analysis for smear layer removal which does not allow this longitudinal observation of dentinal morphology compromise the credibility of the results. (Oral Surg Oral Med Oral Pathol Oral Radiol Endod 2011;112:531-543).

- How was debris and smear layer distinguished between each other?

- The authors did not standardize the area for smear layer removal assessment, which is critically necessary. Other methodologies such as grooves into canals could have provided an improved standardization.

*Results:

- How was normality of data evaluated?

- How was inter-observer bias eliminated?

- Mention in detail the results of inter-group comparisons at different thirds of the root canal system with “P” values.

                                          ******************************

Author Response

This manuscript constitutes an attempt to evaluate the efficacy of three different irrigating systems in removal of intra canal smear layer by conducting an ex-vivo study. The study is relevant to the journal. However, there is no novelty in this study. Previous similar studies have been published in endodontic literature (J Endod 2015;41:1660–1666). Also, the methodology employed is not appropriate. Manuscript is very poorly written. English needs to be improved significantly. Some of the specific queries have been addressed below.

We appreciated the time and effort that the reviewer has dedicated to providing his valuable feedback on our manuscript. Below we report the reply point by point to her/his comments and concerns. The changes have been highlighted in red within the Ms.

Introduction:

  1. Mention previous studies on the efficacy of passive ultrasonics and Endovac in removal of smear layer from root canal system.

Introduction was improved adding previous studies on the efficacy of passive ultrasonics and Endovac in removal of smear layer from root canal system (see lines 73-86).

  1. Mention the hypothesis of the study tested.

The null hypothesis was mentioned, as in lines 85-86.

Methodology:

  1. How was the sample size estimated?

Sample size was based on qualitative variables. Moreover, the sample size used was the one calculated for previous studies [30,38] see lines 193-194.

  1. What was the age groups of patients from whom the teeth were collected? This is one of the very important confounding factor which can influence the results because, in elder patients, the apical portion of the root canal system will be calcified without any open dentinal tubules. This may skew the results when the samples are examined under SEM.

The Reviewer raised an important point. We agree with the Reviewer about the importance of the age of patients from whom the teeth were extracted, indeed the forty human teeth came from subjects aged between 18 and 35 years and had the apex formed. This point has been inserted in Materials and Methods section (line 89).

  1. Authors should provide detailed information regarding how samples were selected, anatomically matched and how groups were formed – authors stated that “single-rooted human teeth” were used. But which single-root teeth? Which canal anatomy? Circular, oval? Nowadays there are several methods that allows authors have similar anatomy and this has to be perfomed otherwise the results are not valid.

As described in the Materials and Methods section, a total of 40 sound single-rooted human teeth were used. These collected teeth were sound, single-canal lower premolars, extracted for orthodontic or periodontal reasons from individuals aged 18 to 35 years. On the basis of the radiographic evaluation, the dental elements with anatomy responding to Vertucci's classification in class 1 and as similar to each other as possible were selected. This point has been inserted in the Materials and Methods section (see lines 93-101). Furthermore, although the anatomy of the root canal (circular and/or oval) may condition the mechanical instrumentation chosen, it does not affect the action of the irrigating solution which acts on the entire surface of the canal. It is important to emphasize that today there are other better methods to evaluate the anatomy and therefore to select samples more similar to each other, despite the traditional radiographic evaluation, such as for example the micro-ct. This limitation was reported in the discussion section (see line 314-317).

  1. Mention the ethical approval number of the project.

Regarding the ethics committee approval, we followed the guidelines set by the Local Ethical Committee of Ospedali Riuniti di Ancona and in agreement with the Helsinki Declaration. In particular, the calcified tissue used for this specific in vitro study was a discard of routinely performed surgical procedures at our Section of Stomatology, DiSCO Department, at the Università Politecnica delle Marche. As we specified in the Ms, written informed consent was obtained from the subjects. Please see others published article in which we used extracted teeth :

- Tosco V, Vitiello F, Furlani M, Gatto ML, Monterubbianesi R, Giuliani A, et al. Microleakage Analysis of Different Bulk-Filling Techniques for Class II Restorations: µ-CT, SEM and EDS Evaluations. Materials. 2020;14(1):31.

- Orilisi G, Monterubbianesi R, Notarstefano V, Tosco V, Vitiello F, Giuliani G, et al. New insights from Raman MicroSpectroscopy and Scanning Electron Microscopy on the microstructure and chemical composition of vestibular and lingual surfaces in permanent and deciduous human teeth. Spectrochim Acta A Mol Biomol Spectrosc. 2021;260:119966.

  1. How was exclusion criteria’s evaluated?

Exclusion criteria were: the presence of root caries, fractures or cracks, previous endodontic treatment, and only teeth with an intact and mature apex and roots longer than 15 mm were selected. Exclusion criteria were detected using a 10x objective (Eclipse Ni, Nikon, Amstelveen, Netherlands). Furthermore, the teeth were visualized by buccolingual and mesiodistal X-ray analysis and those with apical curvature greater than 30° (according to the Schneider method [33]) or calcified root canals were excluded.

This point has been inserted in the Materials and Methods section (lines 93-99).

  1. The volume of the irrigant should be mentioned as “mL” and not “cl”.

We have corrected accordingly, thanks.

  1. It is mentioned that, samples were allocated to three groups (n=10). However, four groups A, B, C, D are mentioned. Clarify!

The Ms was corrected, see line 124-125.

  1. Randomization should have been performed during the allocation of the samples into different groups to avoid bias.

As we wrote in Ms, samples were arbitrarily allocated into four groups (n=10) according to the irrigation system (at lines 124-125)

  1. In groups A and B, what was the time duration of irrigation with NaOCl and EDTA?

The irrigant solution was delivery using at a rate of 1ml/min. The teeth were rinsed with 3 ml of physiological solution, and washed with 3 ml of liquid EDTA at 17% for 1 min.

  1. Why two different types of needle irrigation was performed (Z Rinse and Irrifex)?

Although these two products are both needle for irrigation, they possess different characteristics. Irriflex is a new polypropylene needle just launched on the market which has two side vents at the tip, located back-to-back. Instead, Z-Rinse can be classified as conventional needle, but with two vents on lateral side. Since both of them have not yet been studied, we wanted to investigate their performance so that the results obtained could be useful to evaluate their efficacy.

  1. Mention the manufacturer details of EDTA.

The manufacturer details have been added as follows: EDTA (OGNA Lab S.r.l., Muggiò, MB, Italy).

  1. In group C, what was the time period of NaOCl irrigation? Also, it is not clear in between every cycle of ultrasonic activation, NaOCl was used or not. Mention in detail.

The entire ultrasonic activation cycle consisted of 10 cycles (30s each), and, in each step, the solution and debris were sucked in and replaced with new irrigants (1 mL NaOCL each).

  1. What was the power setting of ultrasonic device used?

The power setting of ultrasonic device was at the maximum power of 45 KHz (± 5 KHz).

  1. In group D, mention the time duration of irrigation with NaOCl and EDTA.

The time duration of Group D irrigation technique was added in the text (At lines 145-153).

  1. In SEM analysis, why the specimens were finished and polished after longitudinal sectioning of the root? What was the rationale behind this?

Samples were finished and polished after longitudinal sectioning of the root in order to obtain a superficial flat surface which was used to lightly groove the three thirds of the root (coronal, middle and apical regions) to standardize SEM analysis. Since the Reviewer also pointed out this aspect in a further question, it was included in the Ms (at lines 167-169).

  1. Knowing that our instruments touch only 50% of the canal walls (J Endod 2011; 37:1394–7; J Endod 2015;41:1545– 50), what is the guarantee that the areas imaged in SEM had a smear layer to begin with?

We thank the Reviewer for this valuable comment. Starting from the assumption that the smear layer is produced by the action of the rotating instrumentation, consequently in the areas where the instrument has not worked due to the anatomy of the canal, there is no smear layer. And since our investigation is focused on the ability to remove the smear layer of different irrigation techniques, we evaluated the area where the residual smear layer is still present along the entire canal.

Furthermore, the Z4 file (25 gauge in D0) worked completely in the apical area and the smear layer is present, as the images obtained demonstrate.

  1. The methodology used to assess smear layer using SEM model is not a sound, reproducible and valid one. Ideally, a longitudinal observation of the canal using micro-CT is regarded a fundamental requirement to study the smear layer removal procedures. Thus, the scanning electron microscopic analysis for smear layer removal which does not allow this longitudinal observation of dentinal morphology compromise the credibility of the results. (Oral Surg Oral Med Oral Pathol Oral Radiol Endod 2011;112:531-543).

We agree with the Reviewer and we added this point as a limitation of the study in the Discussion section. However, there are several studies (cited in the Ms) that investigated the smear layer removal capacity of different irrigation techniques through SEM analysis.

  1. How was debris and smear layer distinguished between each other?

Noteworthy is that the smear layer and debris cannot be differentiated from each other with any degree of certainty in traditional SEM studies, usually using the evaluation of split single-root teeth. However, we wanted to evaluate the presence of a residual amorphous mass known as “smear layer”, which forms and covers the canal walls during mechanical instrumentation. This layer contains inorganic and organic substances including fragments of odontoblastic processes, microorganisms and necrotic materials. Indeed, according to Mader et al. (Mader CL, Baumgartner JC, Peters DD. Scanning electron microscopic investigation of the smeared layer on root canal walls. J Endod 1984;10:477-83.) the smear layer consists of a superficial layer on the surface of the canal wall approximately 1-2 µm thick and a deeper layer inserted into the dentinal tubules to a depth of 40 µm. Smear layer components can be pushed into the dentinal tubules at varying distances. For this reason, we have considered the presence of the smear layer outside and inside the dentinal tubules, according to Hullsman et al. criteria [37].

  1. The authors did not standardize the area for smear layer removal assessment, which is critically necessary. Other methodologies such as grooves into canals could have provided an improved standardization.

As mentioned before, this methodology was described in Materials and Methods section, at lines 167-169.

Results:

  1. How was normality of data evaluated?

Normally, the normality of the data was evaluated with the Shapiro Test. However, since we are used qualitative score we tested the normality with z-test.

  1. How was inter-observer bias eliminated?

Two independent examiners, trained in the scoring process and with concordance verified

with the κ test, scored the samples following the criteria described by Torabinejad et al. [Torabinejad, M.; Khademi, A.A.; Babagoli, J.; Cho, Y.; Johnson, W.B.; Bozhilov, K.; Kim, J.; Shabahang, S. A new solution for the removal of the smear layer. J. Endod. 2003, 29, 170–175.]

  1. Mention in detail the results of inter-group comparisons at different thirds of the root canal system with “P” values.

Table 2 was added to better describe intergroup and intragroup comparison. Letters and numbers were used to specify a statistical difference with p<0.05.

Reviewer 3 Report

please see the file 

Author Response

Comments to authors:

This paper “Evaluation of the Efficacy of Three Different Irrigation Systems on the Removal

of Root Canal Smear Layer: A Scanning Electron Microscopic Study”. I found the work to be

severely lacking in content and have recommended major corrected. My specific comments

are included below, in no particular sequence.

Suggestion to authors

  1. The paper needs improvement in its structure, the background is long. It also needs to be stronger in the argument it is presenting.

The entire Ms has been revised and edited according to the Reviewer`s comment. The Introduction section was shortened, and the novelty of the study was highlighted.

  1. Materials and methods need rewritten.

Materials and Methods have been rewritten clarifying the methodologies used.

  1. A schematic diagram of the experimental setup is required for methodology.

A schematic diagram of the experimental setup was added in the manuscript (see Figure 1).

  1. The work's limitations and implications should be discussed more clearly.

Limitations and implications were discussed in the Discussion section (at lines 314-317)

  1. The result and discussion were poorly written.

Results and discussion were rewritten.

  1. Quality of Figures should be improved.

The resolution of all images was set a 300dpi.

  1. Conclusion: Please revise this part as well as this conclusion part needs to conclude

every result that have been obtained in a precise and understandable manner so that the

viewers could understand well the overall findings that have been obtained.

Conclusion were revised according to the Reviewer comment, in order to highlight in details, the results obtained from this study.

  1. A moderate English revision by a mother tongue reviewer is necessary throughout the

text to remove a few typos and rectify a few grammatical errors that remain in the

manuscript.

For the English revision, a native speaker colleague was asked to edit and modify the entire Ms. Her important help has been acknowledged by the authors.

Round 2

Reviewer 1 Report

Line 99-100- Vertucci's classification in class 1- indicate bibliography

Sample size calculation formula must be explained. In the articles indicated as reference (30 and 38) there is no explanation for sample size calculation. Indeed, the calculation is based on previous studies on similar topic, but a specific formula must be used. Please indicate it.

Table 1 is correctly presented.

Data presented in table 2 is difficult to understand. Give more explanation about correlations (what does a, b, c exactly means? which groups are analysed?), the same with 1, 2, 3.

In discussion section it should be stated if the null hypothesis was validated or not.  

Author Response

  1. Line 99-100- Vertucci's classification in class 1- indicate bibliography

According to your suggestion, the reference was added at line 99 (ref. 33).

  1. Sample size calculation formula must be explained. In the articles indicated as reference (30 and 38) there is no explanation for sample size calculation. Indeed, the calculation is based on previous studies on similar topic, but a specific formula must be used. Please indicate it.

The sample size explanation has been inserted at lines 192-193.

  1. Table 1 is correctly presented.

Thank you for your consideration.

  1. Data presented in table 2 is difficult to understand. Give more explanation about correlations (what does a, b, c exactly means? which groups are analysed?), the same with 1, 2, 3.

The title and legend of Table 2 have been revised to better clarify the meaning of the table and of the superscript letters and numbers.

  1. In discussion section it should be stated if the null hypothesis was validated or not.  

The statement regarding the null hypothesis has been indicated in the discussion section as you suggested (see lines 258-259).

Reviewer 2 Report

Dear Authors,

Thank you for revising the manuscript. Even though you have revised your manuscript substantially, the inherent draw back of your study due to inappropriate methodolgy cannot be eliminated. SEM is no more a valid tool to evaluate the smear layer, since the sole aim of your study was only smear layer analysis.

Author Response

Dear Authors,

Thank you for revising the manuscript. Even though you have revised your manuscript substantially, the inherent draw back of your study due to inappropriate methodology cannot be eliminated. SEM is no more a valid tool to evaluate the smear layer, since the sole aim of your study was only smear layer analysis.

Dear Reviewer,

Thank you for your suggestions which have improved the scientific impact of our work. Nevertheless, all Authors partially agree with your consideration. Although it is true that today there are several analysis tools suitable for the evaluation of the smear layer, it is evident that the SEM analysis allows to visualize in detail the condition of the root canal walls surface after the chemo-mechanical preparation. As you advised, we have highlighted the need to complement this technique used in this work with further analysis and we have included this concept in the limitations of the study. Nevertheless, we believe that this study, like others in the literature using SEM, can be helpful for both researchers and clinicians in understanding and addressing the root canal disinfection procedure which is crucial in endodontics.

Reviewer 3 Report

I recommended that this manuscript be accepted.

Author Response

Thank you for your valuable suggestions and considerations of the manuscript.